# Influence of Meal Sequence and Number of Teeth Present on Nutrient Intake Status: A Cross-Sectional Study

**DOI:** 10.3390/nu15112602

**Published:** 2023-06-01

**Authors:** Sayuri Kida, Norio Aoyama, Toshiya Fujii, Kentaro Taniguchi, Tomomi Yata, Taizo Iwane, Tatsuo Yamamoto, Katsushi Tamaki, Masato Minabe, Motohiro Komaki

**Affiliations:** 1Department of Periodontology, Kanagawa Dental University, 82 Inaoka-cho, Yokosuka 238-8580, Kanagawa, Japan; kdsyr61@gmail.com (S.K.); t.fujii@kdu.ac.jp (T.F.); k.taniguchi@kdu.ac.jp (K.T.); t.oct.o2m6o@gmail.com (T.Y.); m.komaki@kdu.ac.jp (M.K.); 2Center for Innovation Policy, Graduate School of Health Innovation, Kanagawa University of Human Services, 3-25-10 Tonomachi, Kawasaki-ku, Kawasaki-shi 210-0821, Kanagawa, Japan; t.iwane-c2p@kuhs.ac.jp; 3Department of Dental Sociology, Kanagawa Dental University, 82 Inaoka-cho, Yokosuka 238-8580, Kanagawa, Japan; yamamoto.tatsuo@kdu.ac.jp; 4Department of Functional Recovery of TMJ and Occlusion, Kanagawa Dental University, 82 Inaoka-cho, Yokosuka 238-8580, Kanagawa, Japan; tamaki@kdu.ac.jp; 5Bunkyou Dori Dental Clinic, 2-4-1 Anagawa, Chiba 263-0024, Chiba, Japan; minabe-m@wk9.so-net.ne.jp

**Keywords:** meal sequence, nutrient, number of teeth, type 2 diabetes, saturated fatty acids

## Abstract

Intake of fiber, as well as protein, and lipid preloading help to control postprandial glycemic elevation in people with type 2 diabetes and in healthy individuals. However, there are few studies on the awareness of meal sequence and nutrient intake status that consider oral conditions. This cross-sectional study aimed to determine the effects of meal sequences on nutrient intake status and whether these relationships were related to the number of teeth present. The subjects were recruited from the Medical and Dental Collaboration Center of Kanagawa Dental University Hospital between 2018 and 2021. Medical and dental examinations were performed, and a questionnaire was used to determine whether the diet consisted of vegetables, meat or fish, and carbohydrates in that order. Nutrient intake status was assessed using the brief-type self-administered diet history questionnaire. Data were collected from 238 participants. The group with awareness of meal sequence ingested increased nutrients such as n-3 fatty acids, total dietary fiber, calcium, and vitamin C. Saturated fatty acid intake increased in those with fewer teeth, while it was not significantly related to meal sequence. In conclusion, our results showed that meal sequence was associated with nutrient intake status. In addition, the intake of saturated fatty acids increased when many teeth were lost, regardless of meal sequence.

## 1. Introduction

Adequate nutrition is essential to maintain general health. However, inadequate intake of necessary nutrients and lifestyle-related diseases due to overconsumption have become a problem [1]. The meal sequence of eating vegetables and proteins before carbohydrates is gaining attention as a healthier eating habit for people with type 2 diabetes [2,3]. Observational studies have shown that fiber intake from a variety of plant sources could reduce the incidence of obesity and type 2 diabetes [4,5]. In addition to fiber intake, protein and lipid preloading have been found to control postprandial glycemic elevation in people with type 2 diabetes and in healthy individuals by promoting insulin, glucagon-like peptide-1 (GLP-1), and gastric inhibitory peptide secretion [6]. In particular, ingestion of a small amount of protein in advance of a carbohydrate meal has been shown to be effective to modulate gastrointestinal hormone secretion and slow gastric emptying, thereby, improving the subsequent glycemic response to carbohydrates in people with type 2 diabetes. This strategy has the potential to further enhance glucose lowering by dipeptidyl peptidase-4 inhibitors [7]. Males who have lost five or more teeth may have a reduced intake of many nutrients compared with those who have not lost teeth, which may be associated with an increased risk of developing chronic diseases [8], and it has been reported that females with fewer teeth may have an increased risk of cardiovascular disease due to an unhealthy diet [9]. Diet may partially explain the association between oral health and systemic disease. It is clear that the loss of teeth can lead to poor nutrient intake [10]. Poor oral health may cause oral discomfort in the elderly, especially in masticating those foods with fibrous, heterogeneous, and grainy textures [11,12]. Periodontal disease is a common oral disease and a major cause of tooth loss. Fruits and vegetables are significant sources of micronutrients and antioxidants, which have a positive and protective effect on periodontal health [13]. Research results have also shown that a periodontitis group consumed lower amounts of hard foods than a non-periodontitis group, and it was reportedly associated with the nutrient intake of the individuals [14,15]. Therefore, there is a relationship between oral function and nutrient intake status.

Relationships among nutrient intake status, systemic health status, and oral function have been clarified to some extent. However, few studies have investigated oral status in relation to nutrient intake status and meal sequence. This cross-sectional study aimed to determine the effects of meal sequence on nutrient intake status and whether these relationships were related to the number of teeth present.

## 2. Materials and Methods

### 2.1. Study Population

Study participants were recruited from patients who visited the Kanagawa Dental University Hospital, Medical and Dental Collaboration Center between 2018 and 2021. A total of 238 patients consented and participated in the study. Inclusion criteria were age ≥20 years. The Ethics Committee of the School of Dentistry at Kanagawa Dental University approved the present study (approval no. 801). The protocol of this study followed the 1975 Declaration of Helsinki, as revised in 2013. All study participants were informed about the aims and methods of the study prior to participation, and written informed consent was obtained.

### 2.2. Clinical Examinations

All examinations were performed at the Medical and Dental Collaboration Center of Kanagawa Dental University Hospital. The following examinations were performed to determine the general condition of the participants. General patient information such as sex and date of birth was obtained from medical records. Information such as height, weight, and body mass index (BMI) were measured using a body composition analyzer (InBody 460, InBody Japan, Tokyo, Japan). Hemoglobin (Hb) A1c levels were measured by collecting peripheral blood samples. 

The following examinations were performed to determine the oral condition of the participants. The number of teeth present was counted, excluding the third molars. The probing pocket depth and bleeding on probing were measured at 6 points per tooth for all teeth. Probing was performed with a manual probe (PCP-UNC 15, Hu-Friedy, Chicago, IL, USA). These tests were performed by two skilled periodontists (N.A. and M.M.). These periodontal examination records, including probing pocket depth and bleeding on probing, were entered into the periodontal inflamed surface area (PISA) calculation format to calculate the PISA value [16]. The PISA indicates the inflammatory surface area of periodontal pockets and is an indicator that specifically expresses the pathophysiology of periodontal disease, particularly, the state of inflammation. Periodontal disease and diabetes have been reported to be interrelated, and it is an easy-to-understand indicator that serves as “common language” in the collaboration between medicine and dentistry.

Masticatory ability was calculated using a GlucoSensor GS-II (G.C. Corporation, Tokyo, Japan) and gummy jelly (UHA-MIKACTO Corporation, Osaka, Japan), according to the manufacturer’s instructions. For the GlucoSensor GS-II, after chewing the glucose-containing gummy “Glucolam” for 20 s, the gummy and saliva in the mouth were spit into a cup, the saliva was put on a sensor chip, the glucose concentration was automatically measured and displayed in about 6 s, and then recorded. The gummy jelly was chewed 30 times, and all chewed material was spit out into a cup, and the granularity of the gummy was judged on a 10-point scale. Tongue pressure was measured using a balloon-type device (TPM-01, JMS Corporation, Hiroshima, Japan). According to the manufacturer’s instructions, the tongue pressure probe set on the tongue pressure device was inserted into the oral cavity, and the maximum tongue pressure was measured by crushing the balloon with the tongue for approximately 7 s. Oral mucosa moisture values were obtained using a Mucus Oral Moisture Checker (Life Co., Ltd., Saitama, Japan). According to the manufacturer’s instructions, the sensor at the tip was placed in a bag with a special cover, and the measurement was performed by placing the sensor part vertically and at a constant pressure (about 200 g) at about 10 mm from the tip of the tongue. Three consecutive measurements were taken, and the median value was recorded as the measured value.

### 2.3. Diet and Nutrient Intake Examination

A questionnaire was used to investigate whether the participants were eating their meals in the order of “vegetable dishes” (including mushrooms and seaweed), “meat and fish dishes”, and “staple foods” (rice, bread, noodles, etc.) [3]. The response options were “agree”, “somewhat agree”, “sometimes”, “somewhat disagree”, and “disagree”.

Because the number of participants in each response group was small, the participants were reorganized into two groups. The group responding with “agree” and “somewhat agree” was defined as the “awareness of meal sequence” group. The group responding with “sometimes”, “somewhat agree”, and “disagree” was defined as the “unawareness of meal sequence” group.

The brief-type self-administered diet history questionnaire (BDHQ) was used to examine the intake of nutrients [17]. The BDHQ provides a relatively simple, individualized way to determine the amount of nutrients that people living in Japan habitually consume from their regular diet. The questionnaire is designed to obtain information on nutrient intake, food intake, and some other qualitative dietary behavior indicators for each individual. On four A4 sheets of paper, participants completed about 80 questions by themselves in about 15 min. Participants who had difficulty answering the questions were encouraged to answer them with the person who was primarily responsible for meal preparation at home. For elderly participants who had difficulty answering the questions on their own, the interviewer read the questions to them, listened to their answers, and filled in the answers instead. Questionnaires that had already been answered were checked, and any questions that were unclear were asked again for the respondents to correct them. The results were printed out at the support center. Individual result forms were returned to the participants, and feedback was given regarding their dietary habits. Nutrients are expressed per 1000 kcal, a unit of calories, to eliminate the effect of differences in the total amount. 

### 2.4. Statistical Analysis

The normality of the data distribution was confirmed by using the Shapiro–Wilk test. Numerical data with skewed distributions are presented as medians and interquartile ranges. The systemic and nutrient intake statuses of the two groups were compared using the Mann–Whitney U test. The ratio of male to female participants in the two groups was analyzed using a chi-square test. 

In addition, a multiple regression analysis was performed with nutrient intake status as the objective variable, meal sequence and number of teeth present as the explanatory variables, and sex and age as covariates, to confirm the relationships among nutrient intake status, meal sequence, and number of teeth present. Dummy variables were used for the number of teeth present and sex. The number of teeth present was set to 0 for fewer than 20 teeth and 1 for 20 or more teeth. Sex was set to 0 for male and 1 for female. Age was a continuous variable. In Model 1, the explanatory variables were age, sex, and meal sequence. In Model 2, the number of teeth present was added to the explanatory variables of Model 1, i.e., age, sex, and meal sequence, to examine whether the number of teeth present affected the association between nutrients and meal sequence.

The JMP version 14.2.0 software (SAS Institute Inc., Cary, NC, USA) was used for all statistical analyses. All tests were performed considering a significance of *p* < 0.05.

## 3. Results

The participants’ characteristics are listed in Table 1. Among the 238 participants, 177 participants (74%) were aware of their eating order. The group that was conscious of meal sequence was significantly younger than the other group and had a high percentage of females, a low BMI, more teeth, and a good occlusion score. There were no significant differences in the HbA1c level, PISA, occlusal ability, tongue pressure, or oral mucosal moisture between the groups.

Nutrient intake status of the two eating order groups is shown in Table 2. The group with awareness of meal sequence ingested more nutrients, such as lipids, n-6 fatty acids, total dietary fiber, potassium, calcium, iron, vitamin C, folic acid, magnesium, zinc, alpha-tocopherol, vitamin K, vitamin B1, vitamin B2, and vitamin B6. No significant differences were obtained for protein, saturated fatty acids, n-3 fatty acids, phosphorus, retinol, vitamin D, vitamin B12, and salt.

The results of the multiple regression analysis with nutrient intake status as the objective variable, adjusted for sex and age, are shown in Table 3. The explanatory variables in Model 1 were age, sex, and meal sequence. Nutrient intake was significantly higher for lipids, n-3 fatty acids, n-6 fatty acids, total dietary fiber, potassium, calcium, iron, vitamin C, and folic acid, depending on meal sequence. Carbohydrate intake was significantly lower depending on meal sequence. In Model 2, the explanatory variables were age, sex, meal sequence, and number of teeth present. Similar to Model 1, the meal sequence was significantly associated with nutrients such as lipids, n-3 fatty acids, n-6 fatty acids, carbohydrates, total dietary fiber, potassium, calcium, iron, vitamin C, and folic acid, with similar values of non-standardized B, SE, and standardized *β*. Although the number of teeth present was not associated with the intake of most nutrients, saturated fatty acid intake was higher in those with fewer teeth regardless of the meal sequence.

## 4. Discussion

The results of this study show that meal sequence influences nutrient intake status. Considering sex, age, and number of residual teeth, the meal sequence awareness group had higher intakes of fat, n-3 fatty acids, n-6 fatty acids, total dietary fiber, potassium, calcium, iron, vitamin C, and folic acid. Carbohydrate intake was higher in the meal sequence unawareness group. These associations were not affected by the number of teeth present. Furthermore, those with 20 or fewer teeth showed a higher intake of saturated fatty acids, regardless of their meal sequence. The present results suggest that meal sequence may improve nutrient intake status.

Glucose area and glucose peak were lower when carbohydrates were taken at the end of the meal compared to when carbohydrates were taken at the beginning of the meal [18]. Intake of foods in the order of vegetables, meat or fish, and staple foods has been shown to control postprandial plasma glucose elevation in people with type 2 diabetes and in healthy individuals. This is due to the fact that, in addition to fiber intake, protein and lipids promote the secretion of insulin, GLP-1, and gastric inhibitory peptide [4,5,6]. Fiber, one of the crunchy foods, is associated with improved insulin sensitivity and cardiovascular disease, colon health, bowel motility, and multiple medical conditions. Fiber intake also correlates with mortality, which was indicated by a report showing that fiber intake was associated with mortality rate and change in overall lifestyle-related disease [19]. However, most people in the Western world reportedly consume less than one-third of the recommended level of dietary fiber [20], and even in Japan, there is a lack of dietary fiber and other necessary nutrients [1]. For this reason, there is a worldwide need for acquisition of desirable dietary habits based on guidance for proper dietary intake.

This study was designed for participants aged 20 years and older; however, the majority of participants were aged 60 years and older. Elderly people with fewer functional dental units tend to have difficulty in occlusion and swallowing, and therefore, avoid stringy (including meat), crunchy (including vegetables), and dry solid (including breads) foods [21]. This also suggests that participants with fewer teeth consume less vegetables, fish, and crustaceans, and that there is a significant relationship between the intake of nutrients such as minerals and vitamins from food and tooth loss [22]. Thus, it is possible that those with fewer teeth are more concerned about choosing readily available foods than being aware of nutritional balance and the meal sequence in which they regularly eat. Therefore, it is recommended that people with fewer teeth restore masticatory function through prosthetic dental treatment. Furthermore, it is important for those who provide dietary guidance to pay attention to the condition of the oral cavity and the number of teeth of the subject, and to provide dietary guidance accordingly.

A well-balanced diet is essential to maintain good health. In the present study, the group that was more aware of the meal sequence consumed more nutrients (Table 2). BDHQ, a nutrient intake status survey, was used in this study because it provided satisfactory ranking ability for the energy-adjusted intake of many nutrients among the present Japanese population and for estimating the mean values of micronutrients. It is a relatively simple questionnaire that examines, on an individual basis, the amount of nutrients habitually consumed from regular foods (excluding supplements, etc.) by adults living in Japan. It provides information on nutrient intake status, food intake, and some other qualitative dietary behavior indicators for each individual. It consists of about 80 questions and calculates the nutrient intake for 58 foods and more than 100 nutrients. In addition, respondents receive an individual results form (result sheet) to provide feedback on the status of their eating habits [17]. The use of the BDHQ, in this study, made it possible to examine the relationships among a number of nutrients and dietary attitudes and the current number of teeth present.

Participants who consumed an anti-inflammatory diet had more teeth present than those who consumed a proinflammatory diet, and it has been reported that adherence to an anti-inflammatory diet was associated with fewer missing teeth [23]. Moreover, poor-quality diet has been shown to be associated with the development of periodontal disease [24]. In individuals with periodontitis, the use of omega-3 fatty acid dietary supplements as an adjunct to nonsurgical periodontal therapy has resulted in improved periodontal tissue compared with nonsurgical periodontal therapy alone [25]. It has been reported that dietary calcium intake was important for maintaining alveolar bone, and that the mineral intake ratio may affect metabolism [26]. A diet rich in omega-3 fatty acids, vitamins C and D, and fiber as well as low in carbohydrates has been shown to significantly reduce inflammation of periodontal tissue around teeth [27,28]. Nutritional factors have tended to influence current inflammatory pathophysiology of periodontal tissue, such as bleeding on probing and PISA [29]. Thus, it has been suggested that nutrition may be related to oral condition as well as general health.

It has been reported that simple dietary advice after dental prosthetic treatment can improve nutrient intake status [30]. Providing nutritional advice can contribute to effective dental treatment. Therefore, the role of dentistry is not only to restore masticatory ability through periodontal and prosthodontic treatment, but also to continue to provide nutritional support once masticatory ability has been restored. Dentistry is a medical institution that can be involved with patients over the long term through regular dental maintenance. Dental hygienists in charge of dental maintenance are expected to have the knowledge to provide effective nutritional advice. In addition, there is a strong correlation between systemic and periodontal diseases. Medical and dental institutions should, therefore, work together to address lifestyle-related diseases.

Interventions in a meal sequence, which was the focus of this study, have been reported to be more easily followed and adhered to than interventions in a nutritional balance program [31,32]. Eating vegetables before “meat and rice” greatly improves glucose excursion, and interventions for the meal sequence facilitate GLP-1 secretion, which inhibits appetite and potentially leads to long-term diabetes and obesity prevention, indicating the benefits of consuming protein, fat, and fiber before carbohydrates [30]. The secretion of GLP-1 following small intestinal nutrient exposure differs substantially between females and males [33]. The difference in GLP-1 response also predicts gastric emptying [34], which is a major determinant of appetite and postprandial glycemic and is often accelerated in type 2 diabetic patients without complications [35]. In contrast, a systematic review that investigated the effectiveness of the carbohydrate-later meal pattern for people with type 2 diabetes mellitus reported that small differences in hemoglobin A1c may occur, as well as small differences in plasma glucose, insulin, and incretin, at 120 min postprandial. However, there is little evidence for the potential effectiveness of recommending carbohydrates in postprandial dietary patterns beyond standard dietary advice for type 2 diabetic patients and the need for further large-scale studies [36]. This study was conducted because meal sequences are currently a hot topic of discussion. Meal sequence is one of the dietary recommendations for diabetics. Diabetics as well as a small number of healthy people are aware of meal sequence to prevent postprandial hyperglycemia and obesity. However, this is not a concept that has permeated the entire population, and it is still unclear to what extent awareness of meal sequence has an effect on health. Therefore, we conducted this study to evaluate the extent to which concern for meal sequence affects nutrient intake status.

A multiple regression analysis with nutrient intake status as the objective variable and age, sex, meal sequence, and number of teeth present as explanatory variables was performed in this study. The ratio of elderly females in Japan who tend to be undernourished has been reported to have increased significantly over the past decade [37]. Since age and sex have been shown to influence nutrient intake status, age and sex were used as explanatory variables. It was showed that meal sequence was independently associated with the intake of nutrients, such as fat, n-3 fatty acids, carbohydrates, dietary fiber, potassium, calcium, iron, vitamin C, and folic acid (Table 3). Although the present study found only a small association between the number of teeth present and nutrient intake status, those with a few teeth showed a higher intake of saturated fatty acids regardless of their meal sequence. It was consistent with a previous study which showed that tooth loss in adults was associated with poor diet quality and reduced intake of most nutrients [38]. The participants in this study might have been relatively health conscious with well-controlled BMI and HbA1c, and been likely to have a high awareness of nutrient intake.

This study showed, for the first time, that awareness of meal sequence may improve nutrient intake status. The results suggest that nutritional guidance based on meal sequence can be provided not only during medical treatment but also during dental treatment. Periodontal disease is correlated with diabetes, and nutrition has been shown to be associated with inflammation of periodontal tissue [13]; advising on the order in which to eat may help in the treatment of periodontal disease. In addition, the multiple regression analysis showed that saturated fatty acid intake was higher when the number of teeth present was lower, regardless of the awareness regarding the meal sequence (Table 3). This is consistent with reports that the elimination of certain fruits and vegetables due to oral health problems increased the consumption of high-fat and sugary foods [39]. Fewer number of teeth present, reduced food intake, and lower nutrient intake status increase the risk that elderly will require nursing care. Therefore, nutritional guidance and dental maintenance to prevent tooth loss, as well as dental treatment and sustainable nutritional guidance for those with fewer teeth, are desirable. It has also been reported that an increased intake of dietary saturated fatty acids was significantly associated with a high number of periodontal events in non-smokers [40]. Intake of total fatty acids and saturated fatty acid has been reported to be significantly associated with serum total and low-density lipoprotein cholesterol [41]. On the one hand, these findings support the current dietary recommendations to replace saturated and trans fats with unsaturated fats [42], and questions regarding the overconsumption of saturated fatty acids are being raised. On the other hand, dietary recommendations to reduce intake without considering specific saturated fatty acids or food sources are not consistent with the current evidence base. Reports have indicated that restricting intake of saturated fatty acids reduced intake of nutrient-dense foods (e.g., dairy products and unprocessed meats), which helped to improve malnutrition, deficiencies, and frailty [43]. There are also reports that the content of dietary saturated fatty acid intake should be considered in nutritional guidance [44]. Therefore, further research on saturated fatty acid intake is needed.

The limitations of this study should be acknowledged. This was a cross-sectional study; thus, we can only hypothesize an association among nutrient intake status, meal sequence, and number of teeth present. The socioeconomic status of the participants and whether they had received nutritional counseling in the past was not considered. Higher quality diets are, in general, consumed by better educated and more affluent people [45]. Therefore, future studies with additional socioeconomic conditions could have interesting results. The present study was not limited to participants who were visiting the dental hospital for the first time. Thus, the stage of dental care differed among participants. Moreover, sample size calculation was not performed prior to the study.

In conclusion, our results showed that meal sequence was associated with nutrient intake status. In addition, the intake of saturated fatty acids increased when many teeth were lost, regardless of meal sequence.

## Figures and Tables

**Table 1 nutrients-15-02602-t001:** Characteristics of participants in the two meal sequence groups.

	Awareness	Unawareness	
Variable	*n* = 177	*n* = 61	*p*
Age	68 (60.0–74.5)	74 (67.0–80.5)	<0.0001
Sex *	f 123: m 54	f 33: m 28	0.029
BMI (kg/mm^2^)	22.7 (20.3–24.6)	24.2 (20.5–25.5)	0.029
HbA1c (%)	5.7 (5.4–6.0)	5.7 (5.4–6.0)	0.376
Teeth number	25 (23–27)	23 (15–26)	0.001
PISA (mm^2^)	143.4 (74.5–245.2)	159.45 (78.8–297.3)	0.448
Gummy occlusion	6 (5–7)	5 (3–6)	0.002
Occlusal ability	191 (150–235)	180 (149.8–242.8)	0.569
Tongue pressure	32.1 (27.5–37.8)	34.55 (26.3–39.2)	0.745
Oral moisture	26.5 (24.6–27.9)	26.2 (24.6–27.6)	0.512

Mann–Whitney U test. * Pearson’s chi-square test. BMI, body mass index; HbA1c, hemoglobin A1c; PISA, periodontal inflamed surface area.

**Table 2 nutrients-15-02602-t002:** Nutrient intake status of the two meal sequence groups.

	Awareness	Unawareness	
Variables	*n* = 177	*n* = 61	*p*
Protein (%/1day-kcal)	18.53 (16.15–20.76)	17.33 (15.26–20.16)	0.179
Lipid (%/1day-kcal)	31.17 (28.47–34.71)	30.42 (25.72–33.08)	0.037
Saturated fatty acid (%/1day-kcal)	8.21 (7.21–9.63)	8.21 (7.19–9.13)	0.426
n-3 Fatty acids (g/1000 kcal/day)	1.85 (1.54–2.24)	1.71 (1.33–2.20)	0.086
n-6 Fatty acids (g/1000 kcal/day)	6.48 (5.75–7.40)	6.07 (5.20–6.99)	0.013
Carbohydrates (%/1day-kcal)	50.00 (45.78–54.69)	53.35 (47.81–57.39)	0.024
Total dietary fiber (g/day)	8.11 (6.74–9.57)	7.29 (5.62–8.83)	0.004
Potassium (mg/1000 kcal/day)	1762.3 (1515.8–2081.0)	1589.0 (1250.6–1863.4)	0.001
Calcium (mg/1000 kcal/day/)	392.0 (315.6–467.6)	363.2 (294.0–409.3)	0.014
Iron (mg/1000 kcal/day)	5.35 (4.63–6.28)	5.02 (4.19–5.85)	0.008
Vitamin C (mg/1000 kcal/day)	87.97 (67.58–112.73)	71.86 (44.96–102.78)	0.003
Folic acid (μg/1000 kcal/day)	246.9 (200.3–293.0)	218.5 (155.2–262.2)	0.002
Magnesium (mg/1000 kcal/day)	166.9 (14.5–195.6)	156.3 (131.6–179.1)	0.009
Zinc (mg/1000 kcal/day)	4.99 (4.64–5.48)	4.79 (4.32–5.29)	0.024
Phosphorus (mg/1000 kcal/day)	697.7 (607.9–785.6)	679.3 (583.0–766.1)	0.175
Retinol (µg/1000 kcal/day)	487.9 (350.1–626.3)	447.4 (316.0–583.1)	0.253
Vitamin D (µg/1000 kcal/day)	9.17 (6.59–14.95)	11.95 (7.36–16.08)	0.213
Alpha-tocopherol (mg/1000 kcal/day)	5.09 (4.51–5.77)	4.47 (3.74–5.55)	0.001
Vitamin K (µg/1000 kcal/day)	228.3 (167.3–295.1)	171.2 (111.6–246.1)	0.001
Vitamin B1 (mg/1000 kcal/day)	0.52 (0.46–0.59	0.49 (0.40–0.54)	0.002
Vitamin B2 (mg/1000 kcal/day)	0.94 (0.80–1.06)	0.83 (0.74–0.96)	0.002
Vitamin B6 (mg/1000 kcal/day)	0.87 (0.75–0.99	0.82 (0.67–0.96)	0.033
Vitamin B12 (mg/1000 kcal/day)	6.55 (4.84–9.42)	7.12 (4.68–9.85)	0.708
Salt (g/1000 kcal/day)	6.24 (5.43–7.18)	6.43 (5.58–7.40)	0.506

Mann–Whitney U test.

**Table 3 nutrients-15-02602-t003:** Results of the multiple regression analysis with nutrient intake status as the objective variable and adjusted for sex and age.

		Model 1				Model 2			
		Non-Standardized				Non-Standardized			
Target Variable	Explanatory Variable	B	SE	Standardized β	*p*	B	SE	Standardized β	*p*
Protein (%/1day-kcal)	Meal sequence	0.951	0.511	0.119	0.064	0.949	0.528	0.119	0.073
	Teeth number					0.000	0.042	0.001	0.992
Lipid (%/1day-kcal)	Meal sequence	2.358	0.888	0.176	0.009	2.738	0.912	0.204	0.003
	Teeth number					−0.122	0.072	−0.123	0.091
Saturated fatty acid (%/1day-kcal)	Meal sequence	0.405	0.285	0.095	0.157	0.551	0.292	0.129	0.060
	Teeth number					−0.047	0.023	−0.148	0.043
n-3 Fatty acids (g/1000 kcal/day)	Meal sequence	0.182	0.078	0.152	0.022	0.210	0.081	0.176	0.010
	Teeth number					−0.009	0.006	−0.104	0.147
n-6 Fatty acids (g/1000 kcal/day)	Meal sequence	0.555	0.214	0.173	0.010	0.589	0.221	0.184	0.008
	Teeth number					−0.011	0.017	−0.047	0.523
Carbohydrates (%/1day-kcal)	Meal sequence	−3.309	1.192	−0.180	0.006	−3.688	1.228	−0.201	0.003
	Teeth number					0.122	0.097	0.090	0.211
Total dietary fiber (g/day)	Meal sequence	1.098	0.355	0.197	0.002	1.083	0.367	0.194	0.004
	Teeth number					0.005	0.029	0.012	0.866
Potassium (mg/1000 kcal/day)	Meal sequence	240.703	64.082	0.234	0.000	244.132	66.245	0.238	0.000
	Teeth number					−1.102	5.232	−0.014	0.833
Calcium (mg/1000 kcal/day)	Meal sequence	58.109	17.646	0.205	0.001	63.580	18.185	0.224	0.001
	Teeth number					−1.758	1.436	−0.084	0.222
Iron (mg/1000 kcal/day)	Meal sequence	0.578	0.182	0.200	0.002	0.599	0.188	0.207	0.002
	Teeth number					−0.007	0.015	−0.031	0.651
Vitamin C (mg/1000 kcal/day)	Meal sequence	16.953	4.929	0.215	0.001	17.777	5.091	0.225	0.001
	Teeth number					−0.265	0.402	−0.045	0.511
Folic acid (µg/1000 kcal/day)	Meal sequence	38.638	11.475	0.215	0.001	39.171	11.863	0.218	0.001
	Teeth number					−0.171	0.937	−0.013	0.855

Model 1: explanatory variables were age, sex, and meal sequence. Model 2: explanatory variables were age, sex, meal sequence, and number of teeth present, to examine whether the number of teeth present affected the association between nutrients and meal sequence. Values of non-standardized B, SE, and standardized β for meal sequence were similar in Models 1 and 2, suggesting that the number of teeth present had little effect on the association between nutrients and meal sequence.

## Data Availability

The data presented in this study are available on request from the corresponding author.

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
