# Peer review of "Influence of Meal Sequence and Number of Teeth Present on Nutrient Intake Status: A Cross-Sectional Study"

_nutrients, 2023, doi:10.3390/nu15112602_

Round 1

Reviewer 1 Report

This is a very interesting study relating to a topical area of dietary intake and metabolic health. The findings are of interest and the manuscript in general well written. The manuscript could be further improved by addressing the following points:

1. The analytic model included only limited factors for adjustment. It would be of interest to know the socioeconomic status of subjects categorised into the two groups, which may be of relevance to the study findings, ie. whether meal sequence should be an independent factor.

2. The authors should comment more on the accumulating evidence of nutrient preloads in relation to the management of metabolic disorders. In particular, ingestion of a small amount of protein in advance of a carbohydrate meal has been shown to be effective to modulate gastrointestinal hormone secretion and slow gastric emptying thereby improving the subsequent glycaemic response to carbohydrate in people with type 2 diabetes. This strategy has the potential to further enhance glucose-lowering by the DPP-4 inhibitors (PMID: 26786576). 

3. The authors may comment further on the potential sex disparities on the study outcomes. For example, the secretion of GLP-1 following small intestinal nutrient exposure differs substantially between females and males (PMID: 36987568). The difference in GLP-1 response also predicts gastric emptying (PMID: 34791325), which is a major determinant of appetite and postprandial glycaemia and is often accelerated in type 2 diabetic patients without complications (PMID: 30933282). 

Author Response

Dear Reviewer 1,

Thank you very much for your kind review.  Following is our reply to your comments.

Reviewer: 1

Comments and Suggestions for Authors

This is a very interesting study relating to a topical area of dietary intake and metabolic health. The findings are of interest and the manuscript in general well written. The manuscript could be further improved by addressing the following points:

  1. The analytic model included only limited factors for adjustment. It would be of interest to know the socioeconomic status of subjects categorised into the two groups, which may be of relevance to the study findings, ie. whether meal sequence should be an independent factor.

Thank you very much for your comment. We added information in the Limitation.

Higher-quality diets are, in general, consumed by better educated and more affluent people [45]. Therefore, future studies with additional socioeconomic conditions could have interesting results.

  1. The authors should comment more on the accumulating evidence of nutrient preloads in relation to the management of metabolic disorders. In particular, ingestion of a small amount of protein in advance of a carbohydrate meal has been shown to be effective to modulate gastrointestinal hormone secretion and slow gastric emptying thereby improving the subsequent glycaemic response to carbohydrate in people with type 2 diabetes. This strategy has the potential to further enhance glucose-lowering by the DPP-4 inhibitors (PMID: 26786576).

We added information in the Introduction.

In particular, ingestion of a small amount of protein in advance of a carbohydrate meal has been shown to be effective to modulate gastrointestinal hormone secretion and slow gastric emptying thereby improving the subsequent glycemic response to carbohydrate in people with type 2 diabetes. This strategy has the potential to further enhance glucose-lowering by the dipeptidyl peptidase-4 inhibitors [7].

  1. The authors may comment further on the potential sex disparities on the study outcomes. For example, the secretion of GLP-1 following small intestinal nutrient exposure differs substantially between females and males (PMID: 36987568). The difference in GLP-1 response also predicts gastric emptying (PMID: 34791325), which is a major determinant of appetite and postprandial glycaemia and is often accelerated in type 2 diabetic patients without complications (PMID: 30933282). 

We added information in the Discussion.

The secretion of GLP-1 following small intestinal nutrient exposure differs substantially between females and males [33]. The difference in GLP-1 response also predicts gastric emptying [34], which is a major determinant of appetite and postprandial glycemic and is often accelerated in type 2 diabetic patients without complications [35].

We hope that this revised manuscript is suitable for publication in the nutrients. We deeply appreciate your consideration.

Reviewer 2 Report

The paper is aimed at investigating the impact of the awareness about the importance of following a healthy meal sequence on nutrient intake, also defining the role of teeth number.

The study is properly carried out and confirms evidence already reported including the different behavior between men and women regarding the eating habits, as well as in general the awareness about the importance of balanced nutrition on health status. In addition, the study suggests a link between the awareness of meal sequences and a better choice of nutrients, as well as between the intake of saturated fat and number of teeth.

Some points need to be improved.

- In the Methods section, could the authors please explain the categorization and the value assigned to the variables in the regression?  Is sex: F=0 and M=1? What about age?

- Table 3 is difficult to read and understand. Is it possible to show the data in a more comprehensible way?

- In addition, also in the manuscript, Result section, the presentation of these data are not well explained. The author only stated that for defined nutrients the intake is different between the two groups. Could they explain in the text what kind of differences were found for each group?

- The multiple regression analysis has been carried out taking in account age and sex as explanatory variables. Could the authors provide these data considering the potential relevance of these factors on nutrient intake?

Author Response

Dear Reviewer 2,

Thank you very much for your kind review.  Following is our reply to your comments.

Reviewer: 2

Comments and Suggestions for Authors

The paper is aimed at investigating the impact of the awareness about the importance of following a healthy meal sequence on nutrient intake, also defining the role of teeth number.

The study is properly carried out and confirms evidence already reported including the different behavior between men and women regarding the eating habits, as well as in general the awareness about the importance of balanced nutrition on health status. In addition, the study suggests a link between the awareness of meal sequences and a better choice of nutrients, as well as between the intake of saturated fat and number of teeth.

Some points need to be improved.

- In the Methods section, could the authors please explain the categorization and the value assigned to the variables in the regression?  Is sex: F=0 and M=1? What about age?

Thank you very much for your comment. We added information in the Methods.

Dumm variables were used for the number of teeth present and sex. The number of teeth was set to 0 for fewer than 20 teeth and 1 for 20 or more teeth. Sex was set to 0 for male and 1 for female. Age was a continuous variable. In Model 1, the explanatory variables were age, sex, and meal sequence. In Model 2, the number of teeth present was added to the explanatory variables of Model 1: age, sex, and meal sequence.

- Table 3 is difficult to read and understand. Is it possible to show the data in a more comprehensible way?

We placed all "nutritional units" on the second line, aligned the "number of teeth" lines, and adjusted the overall layout to make it easier for the reader to understand.

- In addition, also in the manuscript, Result section, the presentation of these data are not well explained. The author only stated that for defined nutrients the intake is different between the two groups. Could they explain in the text what kind of differences were found for each group?

We explained in texts the differences in each group.

Nutrient intake was significantly higher for lipids, n-3 fatty acids, n-6 fatty acids, total dietary fiber, potassium, calcium, iron, vitamin C, and folic acid, depending on meal sequence. Carbohydrate intake was significantly lower depending on meal sequence. In Model 2, the explanatory variables were age, sex, meal sequence, and number of teeth present. Similar to Model 1, the meal sequence was significantly associated with nutrients such as lipids, n-3 fatty acids, n-6 fatty acid, carbohydrates, total dietary fiber, potassium, calcium, iron, vitamin C, and folic acid.

- The multiple regression analysis has been carried out taking in account age and sex as explanatory variables. Could the authors provide these data considering the potential relevance of these factors on nutrient intake?

We added information in the Discussions.

The ratio of elderly females in Japan who tend to be undernourished has been reported to have increased significantly over the past decade [37]. Since age and sex have been shown to influence nutritional intake, age and sex were used as explanatory variables.

We hope that this revised manuscript is suitable for publication in the nutrients. We deeply appreciate your consideration.